# Radiolabeling of Platelets with ^99m^Tc-HYNIC-Duramycin for In Vivo Imaging Studies

**DOI:** 10.3390/ijms242317119

**Published:** 2023-12-04

**Authors:** Keresztély Merkel, Dávid Szöllősi, Ildikó Horváth, Bálint Jezsó, Zsolt Baranyai, Krisztián Szigeti, Zoltán Varga, Imre Hegedüs, Parasuraman Padmanabhan, Balázs Gulyás, Ralf Bergmann, Domokos Máthé

**Affiliations:** 1Department of Biophysics and Radiation Biology, Semmelweis University, 1094 Budapest, Hungary; 2Biological Nanochemistry Research Group, Research Centre for Natural Sciences, Institute of Materials and Environmental Chemistry, 1117 Budapest, Hungary; 3Clinic of Surgery, Transplantation and Gastroenterology, Semmelweis University, 1085 Budapest, Hungary; 4Duna Medical Center, 1092 Budapest, Hungary; 5In Vivo Imaging Advanced Core Facility, Hungarian Center of Excellence for Molecular Medicine (HCEMM), 1094 Budapest, Hungary; 6Lee Kong Chian School of Medicine, Nanyang Technological University, Singapore 636921, Singapore; 7CROmed Translational Research Centers, 1094 Budapest, Hungary

**Keywords:** platelet radiolabeling, ^99m^Tc-HYNIC-Duramycin, SPECT

## Abstract

Following the in vivo biodistribution of platelets can contribute to a better understanding of their physiological and pathological roles, and nuclear imaging methods, such as single photon emission tomography (SPECT), provide an excellent method for that. SPECT imaging needs stable labeling of the platelets with a radioisotope. In this study, we report a new method to label platelets with ^99m^Tc, the most frequently used isotope for SPECT in clinical applications. The proposed radiolabeling procedure uses a membrane-binding peptide, duramycin. Our results show that duramycin does not cause significant platelet activation, and radiolabeling can be carried out with a procedure utilizing a simple labeling step followed by a size-exclusion chromatography-based purification step. The in vivo application of the radiolabeled human platelets in mice yielded quantitative biodistribution images of the spleen and liver and no accumulation in the lungs. The performed small-animal SPECT/CT in vivo imaging investigations revealed good in vivo stability of the labeling, which paves the way for further applications of ^99m^Tc-labeled-Duramycin in platelet imaging.

## 1. Introduction

Platelets are key players in hemostasis and thrombus formation [1,2], but recently, they have also been linked to tumor progression and inflammation [3,4]. Using appropriate and simple labeling techniques and molecules, these small blood corpuscular components could be readily amenable to clinical translation for image-based biomarker diagnostics. As proposed earlier, a clinically widely used isotope coupled to an appropriate tagging molecule enabled the imaging detection of other circulating blood elements. Thus, extracellular vesicles labeled using the ^99m^Tc-tricarbonyl complex could play an important role in the early detection of malignancies or inflammaging [5]. The role of platelets in the latter inflammaging process hints to the importance of platelet distribution imaging in different non-clotting-related pathophysiological changes, such as aging, senescence, and vascular endothelial autoimmune processes.

Besides routine platelet counts, the in vivo imaging of radiolabeled platelets also holds valuable diagnostic potential, which was recognized several decades ago. Thrombus formation, intravascular clotting, and different entities of peripheric circulation problems have been imaged in the clinic using radiolabeled platelets. Platelet accumulation in thrombi occurs not just during events associated with coagulation and endothelial damage but, evidently, in intra- and peri-tumoral blood clot sites as well. Hence, our intention to provide a clinically applicable new platelet radiolabeling methodology reflects the importance of new tumor imaging necessities and the need for in vivo platelet distribution imaging using isotopes in general. The first radiolabeling studies on platelets date back to the 1950s [6], and platelet scintigraphy or single photon emission tomography (SPECT) in clinical practice with ^111^In-Oxine (^111^In-8-hydroxyquinoline) and ^99m^Tc-HMPAO (hexamethylpropylene amine oxime) became widespread in the 1980s [7,8,9]. These lipophilic complexes are not specific to platelets; therefore, a laborious purification procedure is required before imaging [10]. To overcome this bottleneck, the focus has turned to platelet-specific labels such as receptor-specific radiolabeled peptides and monoclonal antibodies since the 1990s [11,12]. However, these have not yet fully made their way to the clinics in routine nuclear medical practice. Despite the long history of platelet radiolabeling, ^111^In-Oxine still serves as the gold standard among the numerous compounds developed over the years [13], but Indium-111 is an expensive and not always readily available isotope. During the slow washout from the oxine complex, free ^111^In-ions redistribute to bind transferrin, and the imaging results would show bone marrow uptake, making platelet or free indium ion uptake indistinguishable. As recent tumor-related clinical research has shown, there are still unanswered questions regarding the in vivo biodistribution and trafficking of platelets. Therefore, the study of novel platelet-specific radiopharmaceuticals for easy access in pre-clinical models and not just for thrombosis diagnostics is a current topic.

In this paper, with this background, we report a novel method for a lipid-specific radiolabeling of platelets with ^99m^Tc using duramycin, which is a phosphatidylethanolamine-binding (PE) natural, polycyclic peptide antibiotic [14,15,16]. Duramycin conjugated with hydrazinonicotinamide (HYNIC), a bifunctional complexing agent for ^99m^Tc, has previously been proposed as a novel radiopharmaceutical for apoptosis imaging [17,18,19]. Here, we show that ^99m^Tc-HYNIC-Duramycin can be used for the stable radiolabeling of platelets without significant alteration in platelet activity and demonstrate the applicability of the thus-labeled platelets for in vivo small-animal SPECT/X-ray computed tomography (SPECT/CT) imaging investigations. As platelet membranes contain PE, the mechanism of radiolabeling is a simple membrane anchoring of the complexated ^99m^Tc ion to the platelet membrane by duramycin binding to PE.

The reported use of a simplified platelet radiolabel should prove useful in studying tumor, CNS, and immunological research questions. We therefore aimed at performing and reporting this PE-duramycin-based isotope labeling of platelets in our experiments, presenting our results in the following original figures.

## 2. Results and Discussion

Since the aim of our study was to test a novel radiolabeling procedure for platelets, we chose a separation protocol based on size-exclusion chromatography (or gel filtration, in other words), which ensures the highest platelet purity [20,21,22]. It was also reported previously that gel filtration causes less ultrastructural changes in platelets than washing by centrifugation [23]. In this study, the mesoporous Sepharose CL-2B cross-linked agarose gel was used to purify platelets, which ensures the removal of proteins and other biological nanoparticles smaller than 40,000 MDa (for globular proteins). In practice, this means that this stationary phase can remove even low-density lipoprotein particles (LDL) if present. Figure 1a shows the elution profile of the washed platelets based on absorbance measurements at 405 nm. Since the total volume of the used gravity column was 3.5 mL, the excluded or void volume falls slightly above 1 mL. Accordingly, purified platelet fractions were detected between 1 mL and 2 mL elution volumes, which were pooled and used in further steps.

Next, the size and concentration of purified platelets were determined with a novel microfluidic resistive pulse sensing instrument, which operates based on the Coulter principle [24]. The obtained size distribution is shown in Figure 1b. Since platelets are biconvex discoid (lens-shaped) structures, the “diameter” value indicated by microfluidic resistive pulse sensing (MRPS) cannot be directly translated to the physical dimensions of platelets. On the other hand, the mean value of 2.405 m, obtained from the fitting of a lognormal function to the measured distribution agrees well with the known size of platelets, which is 2 to 3 microns in greatest diameter [25]. The measured platelet concentration was 2.27 × 10^9^ mL^−1^, which corresponds to approximately 20% of the initial platelet count in the concentrate. This yield can be explained mostly by loss due to non-specific adsorption during gel filtration, which was quantified in more detail during the radiochemical purification.

Before radiolabeling, the possible activation of platelets by duramycin was tested. The activation test was performed in a 96-well plate by measuring the absorbance as the function of time at 405 nm. Figure 2 shows the time evaluation of absorbance of the duramycin-treated, the control, and the thrombin-treated samples. While a quick activation can be observed for the sample activated by thrombin, the addition of duramycin does not result in significant activation compared with the control sample. Platelet activation was also assessed by transmission electron microscopy (TEM). Figure 3 shows the TEM images of untreated (a), duramycin-treated (b), and activated platelets (with A23187 ionophore in this case). While the duramycin-treated sample (b) shows a typical morphology of resting platelets (a), the ionophore-treated sample contains degranulated platelets due to activation. In summary, both the 96-well plate aggregometry and the TEM investigations indicate that duramycin does not activate the platelets significantly.

Radiolabeling of purified platelets was performed by simply mixing with ^99m^Tc-HYNIC-Duramycin and incubating for 30 min at 30 °C with shaking. Purification was performed with the same gel filtration procedure that was used for the isolation of platelets. Figure 4a shows the elution profile of the reaction mixture determined by measuring the radioactivity of each 0.5 mL fraction in a dose calibrator. Based on the elution profile, the labeling efficiency defined as the activity of the platelet fractions divided by the total activity of the eluate was 57%. On the other hand, the remaining activity on the column was approx. Two-thirds of the total activity of the sample loaded on the column, which is in line with the low yield of the gel-filtration-based purification procedure observed with MRPS. Taking into account the remaining activity of the column, the overall radiolabeling yield was 19%. Nevertheless, the fractions corresponding to the elution volume from 1 mL to 2 mL contain radiolabeled platelets with a purity above 97%, the specific activity of which is sufficient for small-animal SPECT imaging (100 MBq/mL to 300 MBq/mL).

The plasma stability of the radiolabeling was assessed using 10-fold dilution of the labeled platelets in 0.8 μm filtered plasma, and the mixture was incubated at 37 °C with shaking for up to 3 h. At each time point, platelets were separated from the supernatant via filtration using either a 100 kDa MWCO filter or a 0.45 μm pore-size filter. Labeling stability percentages defined as the ratio of the activity of the platelet fraction and the filtrate are shown in Figure 4b. While >99% plasma stability was observed when determined using the 100 kDa MWCO filter, it dropped to 87% when determined with the 0.45 μm pore-size filter. This observation indicates that ^99m^Tc-HYNIC-Duramycin is not released as a single molecule from the surface of platelets but as attached to membrane fragments or to extracellular vesicles released from platelets during incubation.

The specific activity of the ^99m^Tc-HYNIC-Duramycin-labeled platelets in terms of labeled PE lipids per total number of PE lipids in one platelet can be estimated based on the platelet concentration, the activity concentration, the total phospholipid content, and the ratio of externally accessible PE lipids in platelets. The total phospholipid content of platelets was reported to be 3.5 mg/10^10^ platelets, and 25% of that consists of PE [26]. Calculating with an average molecular weight of 750 g/mol, the molar content of PE in platelets is 4.67 × 10^−16^ mol/platelet or 7 × 10^7^ PE molecules/platelet. However, the externally accessible amount of PE in platelets is only 6.9% of the total PE lipids as measured using 2,4,6-trinitrobenzene sulfonate, which yields 4.8 × 10^6^ labelable PE/platelet [27]. This can be compared to the number of ^99m^Tc-HYNIC-Duramycin molecules per platelet that can be calculated from the specific activity of the purified sample. In a typical experiment, we had 100 MBq to 300 MBq activity in 1 mL, with an estimated platelet concentration of 2 × 10^8^ mL^−1^. Using the A = N·λ formula, where A is the activity of the sample in Bq, N is the number of decaying atoms, and λ is the decay constant defined as λ = ln 2/t_1/2_, in which t_1/2_ is the half-life of ^99m^Tc (6 h), and the number concentration of ^99m^Tc in the labeled platelet sample was approx. 3.1 × 10^12^ mL^−1^ to 9.3 × 10^12^ mL^−1^. These formulas yield 0.54–1.6 × 10^4 99m^Tc-HYNIC-Duramycin molecules per platelet, which means that approximately 0.3% to 1% of the PE molecules on the surface of the platelets are labeled.

The in vivo imaging results of labeled platelets 3.5 h p.i. are shown in Figure 5. Tissue distributions in the liver, spleen, and lungs in the percentage of whole-body radioactivity and in SUV are shown in Figure 5b,c, respectively. High liver uptake is apparent from the unsegmented images, which already shows the in vivo stability of the radiolabeling, as free ^99m^Tc-HYNIC-Duramycin has low tissue uptake and fast clearance from the circulation via the renal–urinary system [17].

However, the use of radiolabeled platelets for the imaging of thrombosis was introduced in the 1970’s [28], and studies reporting the biodistribution of in vitro-labeled platelets in small animals are scarce [29]. The distribution of ^111^In-Tropolone and ^111^In-Oxine-labeled platelets in rabbits was reported by Dewanjee et al., who also observed significant liver uptake. Still, the activity in the blood was the highest after 24 h p.i. for both compounds according to this study [30]. Due to the technical difficulties associated with the in vitro radiolabeling of autologous platelets in a clinical setting, the research focus in the field turned to radiolabeled compounds that specifically bind to platelets in vivo. Among these compounds, a radiolabeled monoclonal antibody to platelets [12,31] and bitistatin, an α_IIb_β_3_ (glycoproteins IIb/IIIa) receptor-specific peptide, should be mentioned [11]. Significant activity in the kidneys is reported in these studies, probably due to the in vivo labeling strategy, which results in the free radiolabeled compound in the circulation which is quickly secreted by the renal–urinary system. Recently, Lee et al. reported the PET/CT imaging of platelets labeled with a novel radioiodinated gold nanoprobe in tumor-bearing mice [32]. However, the distribution of the nanoprobe-labeled platelets in healthy animals is not reported in this study. Lee et al. also found the highest uptake of the labeled platelets in the liver followed by accumulation in the blood and in the lungs, with low activity in the kidneys.

Single-photon emission computed tomography (SPECT) radionuclide imaging can provide effective in clinical diagnostic applications because of its high sensitivity (very small amounts of a radionuclide label are sufficient for good resolution and diagnosis) and its ability to provide quantitative, whole-body imaging [33]. This platelet radiolabeling technique is to be translated to clinical trials for both “traditional” diagnostics of deep-vein or other locality thrombosis and for the use of radiolabeled platelets in whole-body tumor imaging. One possible translation to clinical diagnostics of our herein reported method is the imaging of nasopharyngeal carcinoma via platelet labeling because, in the beginning, this carcinoma is limited to the mucosa. A report describes platelet assemblies or clots around the tumor cell embolies, migrating to metastasize into the neighboring anatomical regions [34].

This technetium-based platelet-selective labeling process might be realized in a more uniform and protocol-based way than In-oxime radiolabeling. However, the most important advantage of our method for further clinical use is that patient and staff dosimetry is significantly lower in any ^99m^Tc-based radiopharmaceutical than for ^111^In-oxime. The same dosimetry consideration applies to the half-life of ^99m^Tc as compared to ^111^In. Therefore, this type of kit-based approach, coupled with also standardizable SEC separation as opposed to centrifugation-based hemolysis-prone cell/PLT whole-blood separations, can eventually be more readily trained and applied in clinical practice. The limitation of our duramycin-based platelet radiolabeling method might be represented in the current SEC purification protocol step. In a clinical application, a more seamless and less time-consuming, optimized version of this SEC purification method may need to be provided. To check the thrombosis-related clinical relevance, further studies are also warranted for the evaluation of experimental thrombus sites in pre-clinical models. However, this and other eventual pre-clinical tumor imaging results pertain to a different, more application-oriented in vivo model imaging study, as the results presented here point to a stand-alone new molecular method of obtaining and imaging stably radiolabeled platelets.

## 3. Materials and Methods

### 3.1. Platelet Purification

Washed human platelet concentrate was purchased from the Hungarian National Blood Transfusion Service (Budapest, Hungary) and was used within 24 h after arrival. A 9 mL platelet concentrate was mixed with a 1 mL ACD-A solution in a plastic tube (Greiner VACUETTE^®^ ACD-A, Greiner Bio-One Ltd., Budapest, Hungary), and a 10 μL prostaglandin E1 solution (100 μg/mL in EtOH, Avanti Polar Lipids, San Francisco, CA, USA) was added. First, platelets were centrifuged at 800× *g* for 15 min at room temperature (Nüve NF800R, RA200 swing-out rotor), and the supernatant was discarded. The pellet was gently resuspended in 1 mL HEPES-buffered Tyrode’s solution (THB, 119 mM NaCl, 5 mM KCl, 25 mM, HEPES pH 7.4, 2 mM CaCl_2_, 2 mM MgCl_2_ 6 g/l Glucose, pH 7.4), and 0.5 mL of the washed platelet concentrate was further purified with size-exclusion chromatography (SEC) using a 3.5 mL gravity column filled with Sepharose CL-2B gel (GE Healthcare, Uppsala, Sweden) equilibrated with THB solution. A total of 0.5 mL fractions were collected, and absorbance at 405 nm was measured with a BioTek Synergy 2 plate reader using 96-well plates [35]. Purified platelets corresponding to the void volume of the column (from 1 mL to 2 mL elution volumes) were combined and used for radiolabeling.

### 3.2. Platelet Counting with Microfluidic Resistive Pulse Sensing

Concentration and size characterization of purified platelets was performed with microfluidic resistive pulse sensing (MRPS) using an nCS1 instrument (Spectradyne LLC, Signal Hill, CA, USA) [24]. Samples were diluted 100-fold in a THB solution containing bovine serum albumin (BSA, Merck Life Science Ltd., Érd, Hungary) at 1 mg/mL. MRPS measurements were performed using factory-calibrated TS-10k cartridges with a measurement range from 1500 nm to 10,000 nm.

### 3.3. Platelet-Activation Test

SEC-purified platelets were diluted 10-fold in THB solution, and duramycin (Merck Life Science Ltd., Érd, Hungary) was added to reach a final concentration of 0.35 μg/mL. Activation with thrombin at a 1 U/mL final concentration was used as a positive control. Mixing was performed in a 96-well plate and quickly placed in a BioTek Synergy 2 plate reader, and absorbance was measured at 405 nm at different time points for one hour [35].

### 3.4. Transmission Electron Microscopy

Untreated platelets as a negative control, platelets treated with duramycin at 0.35 μg/mL concentration, and platelets activated with A23187 Ca-ionophore (at 10 μM final concentration, Merck Life Science) as a positive control were investigated by transmission electron microscopy (TEM). Samples were centrifuged and fixed with a fixative solution (containing 3.2% formaldehyde, 0.32% glutaraldehyde, 1% saccharose, 40 mM CaCl_2_, and 0.1 M sodium-cacodylate) at 4 °C overnight. For post-fixation, we applied 1% osmium tetroxide for 1 h. Next, the platelet pellets were dehydrated by graded ethanol series and then embedded in Spurr low-viscosity epoxy resin (Sigma-Aldrich, Hungary, Budapest, Hungary cat. no. EM0300) according to the recommendation of the manufacturer. Ultrathin sections were made using a Reichert U3 ultramicrotome and collected on Formvar-coated copper grids. Uranyl acetate and Reynolds’ lead citrate were used to counterstain.

The samples were examined using a MORGAGNI 268D transmission electron microscope (FEI, Eindhoven, The Netherlands) equipped with a Quemesa 11 MegaPixel bottom-mounted CCD camera (EMSIS, Münster, Germany).

### 3.5. Radiolabeling with ^99m^Tc-HYNIC-Duramycin

HYNIC-Duramycin kit was purchased from Molecular Targeting Technologies (West Chester, PA, USA). Radiolabeling was conducted following the manufacturer’s instructions. Briefly, 0.5 mL of freshly eluted ^99m^Tc-pertechnetate (1 to 3 GBq eluted from an Ultra-Technekow FM 2.15–43.00 GBq technetium generator (Curium, Petten, The Netherlands)) solution was applied to the glass vial of the kit. The vial was incubated at 80 °C for 20 min to complete the labeling. Next, 0.25 mL of CL-2B-purified platelets in THB was mixed with 0.25 mL ^99m^Tc-HYNIC-Duramycin (0.4 to 1 GBq) and incubated for 30 min at 30 °C with 200 RPM shaking in a neoMix cool thermomixer (Neolab, Heidelberg, Germany). Radiolabeled platelets were separated from free ^99m^Tc-HYNIC-Duramycin by SEC using the same protocol as described in the previous section. The fractions corresponding to the void volume of the column (from 1 mL to 2 mL elution volumes) were pooled and used for testing the stability of the radiolabeling or for in vivo SPECT/CT investigations.

The radiolabeling stability test was performed by mixing 0.25 mL of ^99m^Tc-HYNIC-Duramycin-labeled platelets with 2.25 mL of human plasma (supernatant of platelet concentrate after the 800× *g* centrifugation and filtered with a 0.45 μm syringe filter (VWR International, Debrecen, Hungary)). A total of 100 μL aliquots were placed in LoBind Eppendorf tubes and incubated at 37 °C with 200 RPM shaking in a neoMix cool thermomixer (Neolab, Heidelberg, Germany). At each time point, two separation procedures were performed in parallel with separate platelets from the free ^99m^Tc-HYNIC-Duramycin: the samples were filtered with either 0.45 μm (5000× *g*, 3 min) or 100 kDa MWCO (12,300× *g*, 10 min) centrifuge filters, and the activities of the flow through and retentates were measured in a dose calibrator (ISOMED2010, Nuvia, Rueil-Malmaison, France).

### 3.6. In Vivo SPECT/CT Imaging

Five healthy 6-week-old NMRI Nu/Nu mice (body weight between 22 g and 26 g) bred in the Animal House of Semmelweis University were used for biodistribution studies. Animals were allowed free access to food and water and were kept under humidity-, temperature-, and light-controlled conditions. All procedures were conducted in accordance with the ARRIVE guidelines and the guidelines set by the European Communities Council Directive (86/609 EEC) and approved by the Animal Care and Use Committee of the IEM and Semmelweis University (PE/EA/599-5/2021). A volume of 100 μL to 200 μL radiolabeled platelets in THB with an activity of 24.5 ± 1.6 MBq (Mean ± SD) was administered intravenously into the lateral tail vein. Mice were anesthetized with isoflurane for the whole duration of the imaging. SPECT/CT acquisitions were carried out using a nanoScan SPECT/CT (Mediso, Budapest, Hungary) and started 3.5 h post-injection. SPECT and CT acquisitions were reconstructed using Nucline (Mediso, Hungary). Volumes of interest (VOIs) were manually delineated around selected organs (lungs, liver, and spleen). VOI uptake data are reported in % organ activity/injected dose and in standardized uptake value (SUV), which is defined as the ratio of the tissue radioactivity concentration and the injected activity divided by the body weight.

## 4. Conclusions

In this paper, a novel method for the radiolabeling of platelets for in vivo SPECT/CT imaging using ^99m^Tc-HYNIC-Duramycin is proposed. This method is aimed at possible overall use of platelet imaging with SPECT to further answer platelet distribution-related questions in vascular, tumor, immune, and aging biology. Our intention was not expressly the application of the radiolabeled platelets for any detection of thrombi but rather the report on the labeling and stability and the SPECT imaging possibility of the so-radiolabeled platelets for any biological purpose. Therefore, we need to note the limitations in our methods that need to be further tested in different biological pre-clinical whole-body imaging studies before the clinical translation of the method can be studied for patient diagnostics.

As the known mechanism of radiolabeling occurs, duramycin specifically binds to lipids with a phosphatidyl-ethanolamine head-group; hence, it provides a membrane-specific labeling. The performed activity tests indicated that in the concentration used for radiolabeling, duramycin does not cause platelet aggregation. The in vitro plasma stability of the labeling was found to be sufficient for in vivo studies. The performed small-animal in vivo SPECT/CT investigation revealed a good in vivo stability of the labeled platelets, which adumbrate the possible clinical applicability of the platelet-radiolabeling procedure with ^99m^Tc-HYNIC-Duramycin.

## Figures and Tables

**Figure 1 ijms-24-17119-f001:**
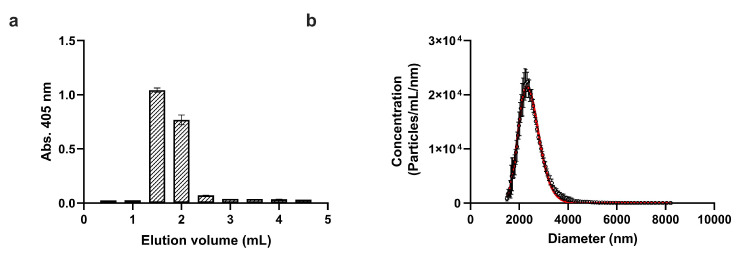
(**a**) Purification of platelets with SEC using a Sepharose CL-2B column. Absorbance of 0.5 mL fractions of the elute was measured in a 96-well plate at 405 nm. Fractions 4 and 5 corresponding to the elution volume from 1 mL to 2 mL containing purified platelets were pooled and used for radiolabeling experiments. (**b**) Particle size distribution of purified platelets as determined via microfluidic resistive pulse sensing (MRPS). Symbols show mean concentration ± SD (n = 3) for each bin with the size of 30 nm, while the solid red line corresponds to the best-fitting lognormal function. The mean diameter (of equivalent spheres) was found at 2.405 μm, while the total concentration of platelets was 2.27 × 10^9^ mL^−1^.

**Figure 2 ijms-24-17119-f002:**
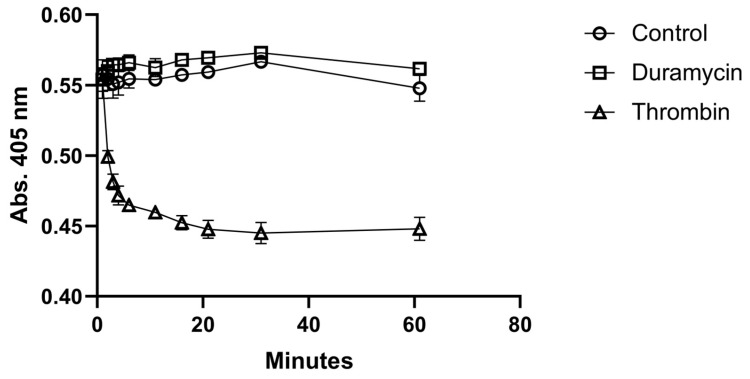
Test of possible platelet activation using duramycin with 96-well plate-based aggregometry. Duramycin was added to the platelets at a 0.35 μg/mL final concentration, which resembles its concentration during radiolabeling. Thrombin at a 1 U/mL final concentration was used as a positive control, while a THB solution was added in the case of the negative control sample.

**Figure 3 ijms-24-17119-f003:**
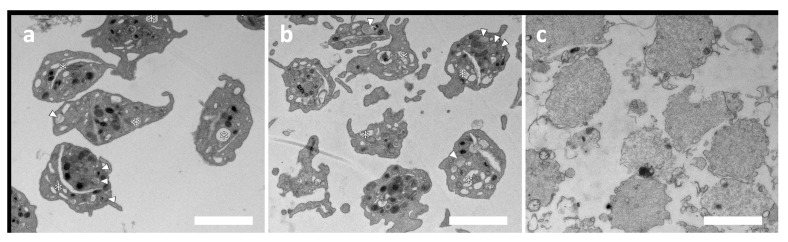
Test of possible platelet activation using duramycin with transmission electron microscopy (TEM). Images of the untreated (**a**) and the duramycin-treated (**b**) samples show the typical morphology of platelets, while the positive control sample (**c**), platelets treated with A23187 ionophore) show degranulated platelets due to activation. The bar represents 2 μm. Asterisks show the open canalicular system, white arrowheads point to alpha-granules in the inactive platelets of panels (**a**,**b**), which are absent in the activated platelets in panel (**c**).

**Figure 4 ijms-24-17119-f004:**
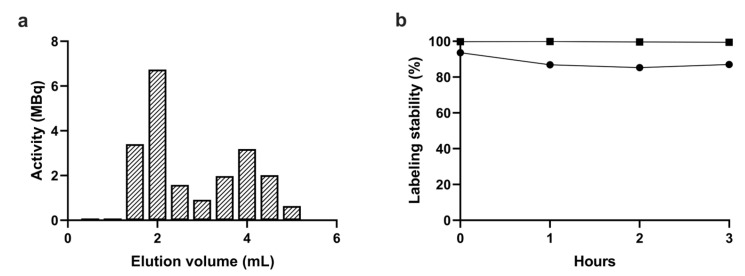
(**a**) Purification of platelets radiolabeled with ^99m^Tc-HYNIC-Duramycin with SEC using a Sepharose CL-2B column. (**b**) Stability test of radiolabeled platelets in plasma using a 100 kDa MWCO filter (squares) and a 0.45 μm filter (circles) to separate platelets from free label.

**Figure 5 ijms-24-17119-f005:**
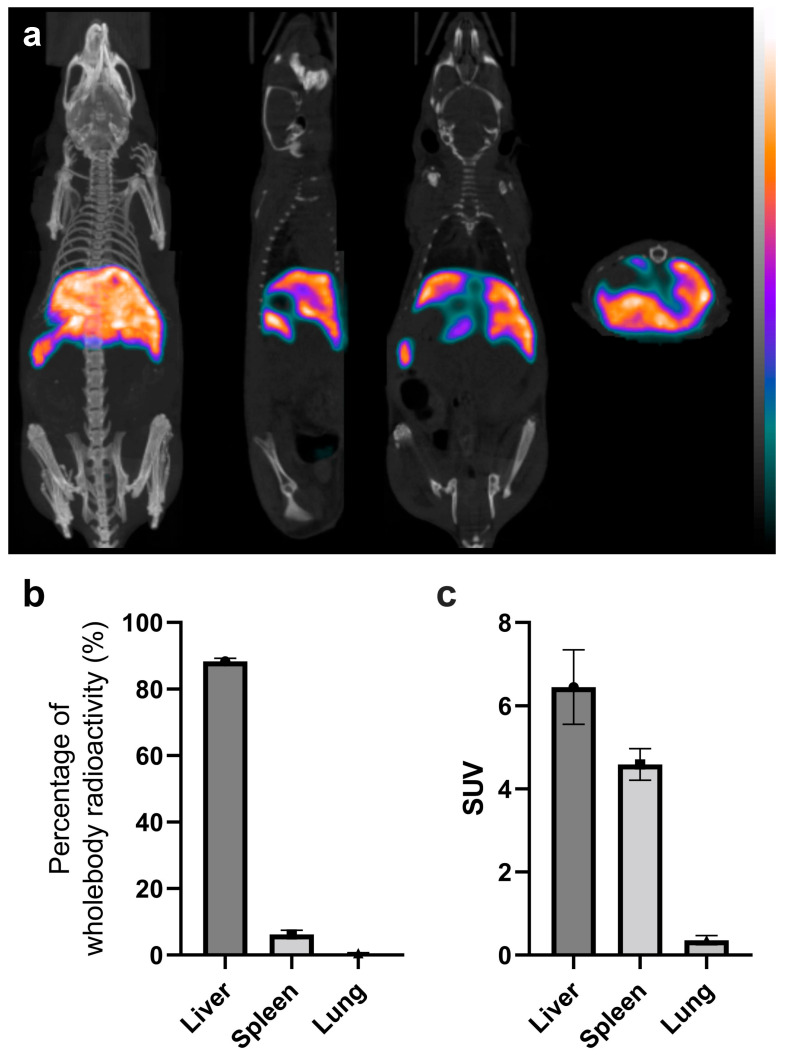
In vivo imaging of ^99m^Tc-HYNIC-Duramycin-labeled platelets. Panel b presents the radioactivity percentage accumulated in the liver, spleen and lungs proportionally to the radioactivity measured in the whole animal body. Panel c shows the standardised uptake values (SUV) of radioactivity in liver, spleen and lungs calculated from the injected radioactivity value. (**a**) In vivo imaging results of labeled platelets 3.5 hours p.i. Tis-sue distributions in the liver, spleen, and lungs in the percentage of whole-body radioac-tivity (**b**) and SUV (**c**).

## Data Availability

Data are available upon reasonable request from the corresponding authors.

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
