# Peer review of "Radiolabeling of Platelets with 99mTc-HYNIC-Duramycin for In Vivo Imaging Studies"

_ijms, 2023, doi:10.3390/ijms242317119_

Round 1

Reviewer 1 Report

Comments and Suggestions for Authors

Dear Authors,

I congratulate you for your interesting manuscript.

However, there are some aspects that require your attention.

1. What is the main question addressed by the research?  In vivo biodistribution of labeled platelets with 99mTc. 2. Do you consider the topic original or relevant in the field? Does it address a specific gap in the field? It underlines a specific gap in the field, because platelets are accumulating at the level of intratumoral necrosis and thrombosis. 3. What does it add to the subject area compared with other published material? This study uses newer SPECT machines, the improvement in SPECT is also helping to a better detection of the marked platelets. 4. What specific improvements should the authors consider regarding the methodology? What further controls should be considered? In the discussions insert a subheading about the limitation of this study. 5. Are the conclusions consistent with the evidence and arguments presented and do they address the main question posed? In the conclusions insert a short sentence about the need to translate these findings from animal studies to human clinical trials. 6. Are the references appropriate? In the discussion section you need to underline possible future use. For example the nasopharyngeal carcinoma could be detected earlier by this technique. Because in the beginning the nasopharyngeal carcinoma is limited to the mucosa and the platelets could clot around the tumor emboly towards the neighboring regions. Reference this to the article by Anghel, I., Anghel, A. G., Dumitru, M., & Soreanu, C. C. (2012). Nasopharyngeal carcinoma -- analysis of risk factors and immunological markers. Chirurgia (Bucharest, Romania : 1990)107(5), 640–645. 7. Please include any additional comments on the tables and figures. Mention that the figures are original.   Looking forward to receiving the improved version of the manuscript.

Author Response

Please see the attached responses.

Reviewer 2 Report

Comments and Suggestions for Authors

A manuscript (ijms-2718050) which is entitled: “Radiolabeling of platelets with 99mTc-HYNIC-Duramycin for in vivo imaging studies” (already published as a non-peer-reviewed article in the internet) has been submitted to the scientific journal Cells.

The study is carefully executed and the results are sound. I am convinced that this draft will be of interest for the general readership of the journal Cells. Nevertheless, I was a bit puzzled why this article of the first author is not cited in the mamiscript:

Varga Z, Gyurkó I, Pálóczi K, Buzás EI, Horváth I, Hegedűs N, Máthé D, Szigeti K. (2016) Radiolabeling of Extracellular Vesicles with (99m)Tc for Quantitative In Vivo Imaging Studies. Cancer Biother Radiopharm.31(5):168-73. doi: 10.1089/cbr.2016.2009. PMID: 27310303.

Furthermore, I would be benefical for the article if thes two publications are discussed in detail:

Gawne PJ, Man F, Blower PJ, T M de Rosales R. (2022) Direct Cell Radiolabeling for in Vivo Cell Tracking with PET and SPECT Imaging. Chem Rev., 122(11):10266-10318. doi: 10.1021/acs.chemrev.1c00767. Epub 2022 May 12. PMID: 35549242; PMCID: PMC9185691.

Clough, A. V.; Audi, S. H.; Haworth, S. T.; Roerig, D. L. (2012) Differential Lung Uptake of 99mTc-Hexamethylpropyleneamine Oxime and 99mTc-Duramycin in the Chronic Hyperoxia Rat. Model. J. Nucl. Med. 53, 1984−1991.

Author Response

Please see the attached responses.
